# Two Sides of The Same Coin: Normal and Tumoral Stem Cells, The Relevance of In Vitro Models and Therapeutic Approaches: The Experience with Zika Virus in Nervous System Development and Glioblastoma Treatment

**DOI:** 10.3390/ijms241713550

**Published:** 2023-08-31

**Authors:** Rosaria Tinnirello, Cinzia Maria Chinnici, Vitale Miceli, Rosalia Busà, Matteo Bulati, Alessia Gallo, Giovanni Zito, Pier Giulio Conaldi, Gioacchin Iannolo

**Affiliations:** 1Department of Research, IRCCS ISMETT (Istituto Mediterraneo per i Trapianti e Terapie ad Alta Specializzazione), Via E. Tricomi 5, 90127 Palermo, Italy; rtinnirello@ismett.edu (R.T.); cchinnici@fondazionerimed.com (C.M.C.); vmiceli@ismett.edu (V.M.); rbusa@ismett.edu (R.B.); mbulati@ismett.edu (M.B.); agallo@ismett.edu (A.G.); gzito@ismett.edu (G.Z.); pgconaldi@ismett.edu (P.G.C.); 2Regenerative Medicine and Immunotherapy Area, Fondazione Ri.MED c/o IRCCS ISMETT, 90127 Palermo, Italy

**Keywords:** neural stem cells, glioblastoma stem cells, cancer stem cells, Zika virus, nervous system development, brain

## Abstract

Neural stem cells (NSCs) were described for the first time more than two decades ago for their ability to differentiate into all neural cell lineages. The isolation of NSCs from adults and embryos was carried out by various laboratories and in different species, from mice to humans. Similarly, no more than two decades ago, cancer stem cells were described. Cancer stem cells, previously identified in hematological malignancies, have now been isolated from several solid tumors (breast, brain, and gastrointestinal compartment). Though the origin of these cells is still unknown, there is a wide consensus about their role in tumor onset, propagation and, in particular, resistance to treatments. Normal and neoplastic neural stem cells share common characteristics, and can thus be considered as two sides of the same coin. This is particularly true in the case of the Zika virus (ZIKV), which has been described as an inhibitor of neural development by specifically targeting NSCs. This understanding prompted us and other groups to evaluate ZIKV action in glioblastoma stem cells (GSCs). The results indicate an oncolytic activity of this virus vs. GSCs, opening potentially new possibilities in glioblastoma treatment.

## 1. Introduction

Since the last century, it has been assumed that the nervous system (NS) cannot replicate or undergo repair processes. The relatively recent study of new cells (e.g., neurons, glia) within the adult NS offers input for the isolation and the analysis of neural precursors/progenitors (NPCs) or neural stem cells (NSCs). Adult NSCs were isolated for the first time from a mouse central nervous system (CNS) in 1992 [1]. These cells grow in suspension, as spherical clusters, in medium without supplemented serum but with epidermal growth factor (EGF) and basic fibroblast growth factor (bFGF). The NSCs can be expanded as neurospheres or driven to differentiation into various nervous system lineages (Figure 1) [1]. Several groups have reported on the isolation and characterization of similar populations from different species, included human. For example, in the adult brain, mammalian NSCs reside in the subventricular zone (SVZ) and subgranular zone (SGZ) of the hippocampus (recently reviewed by Llorente et al., 2022 [2]). These cells reside in their niche, and have a particular ability to resist apoptosis induced by acute brain injury, while exhibiting endogenous mechanisms of resistance [3,4].

The first characterization of tumor stem cells occurred in the hematopoietic system, where a specific population (CD34+, CD38−) was found to be responsible for leukemic propagation [5] in severe combined immunodeficient (SCID) mice. Similarly, brain tumor stem cells (BTSCs) (Figure 2) have been isolated and characterized [6]. This brain tumor cell population shares NSC characteristics and growth conditions. It is generally acknowledged that cancer arises from transformed stem cells, and this theory was recently reviewed by Liang and Kaufmann [7], extending it to the nervous system [8,9]. Several studies have pointed to the characterization of NSCs and BTSCs, both to provide therapeutic solutions for neurodegenerative diseases or traumatic injuries, and to prevent uncontrolled proliferation in neoplastic transformation.

In this review, we seek to summarize some of these achievements, focusing our attention on the thread that binds normal and transformed cells and on the importance of an accurate analysis of their common features. In particular, we have focused on the link between neural development and cancer, as demonstrated by the Zika virus (ZIKV) lesson, in which a virus that impairs neural development can represent a new hope for glioblastoma treatment [10,11,12].

## 2. Neural Stem Cells

NSCs are the precursors that participate in CNS development at the embryonal level, and that provide maintenance and repair to the CNS in adult life. NSCs can undergo self-renewal replication or differentiate into the main neural cell types: neurons, astrocytes, and oligodendrocytes. These cells grow in vitro as neurospheres (Figure 1), and their propagation in culture takes place in absence of serum and in presence of EGF and bFGF [13]. Moreover, they can differentiate into various lineages (e.g., neurons, astrocytes, and oligodendrocytes) in adhesion (Figure 3), in the presence of low serum concentration (1–2%). The mechanisms responsible for the maintenance of an undifferentiated state have not been fully elucidated, though some of the players involved in the neural developmental program have been defined. Among them a key role is ascribed to Notch, a transmembrane receptor involved at various levels and in different species in neurodevelopmental processes [14]. This receptor, originally described in the Drosophila nervous system, consists of a single-pass transmembrane protein that is activated by various ligands (Jagged, Delta, Serrate), triggering, with their binding, a proteolytic cleavage cascade, firstly from a metalloprotease (ADAM10, ADAM17), and subsequently from the γ-secretase complex. Hence, Notch intracellular domain (NCID) is released from the membrane and translocated into the nucleus, where it promotes the transcription of specific genes (HES/HEY family). In addition to the activity crucial for the maintenance of the undifferentiated status in the nervous system, Notch can also determine cell fate during differentiation [15]. In particular, the down-modulation of the Notch receptor or its ligands results in a dramatic reduction of the NSC pool [16,17]. While its constitutive activation from NCID inhibits neuronal differentiation, this action sustains the idea of a permissive role of Notch that contributes to the maintenance of a pluripotent state in precursor cells [18]. This is supported by the observation of Notch involvement in the regulation of cell fate by controlling precursor differentiation [19] through neurogenesis and gliogenesis regulation in the embryonal stages [20].

Notch function in neural stem cell maintenance is counteracted by its historic antagonist: the docking protein Numb [21]. Numb’s role in NS normal and pathological differentiation has been studied for several years (reviewed by Ortega-Campos and García-Heredia [21]). This docking protein is involved in many cellular processes, from clathrin-dependent pit internalization to the modulation of E3 ligases (Itch, MDM2). Its role in counteracting Notch goes back to Drosophila genetic studies [21], though the mechanisms have not been fully elucidated. Numb controls the differentiation processes in nervous system development [21]. One of the mechanisms through which Numb inhibits Notch is mediated by the Itch E3 ligase, the binding of which induces Notch ubiquitination and, subsequently, its proteasomal-dependent degradation. Conversely, the binding to Itch inhibits p73 proteasome-dependent degradation [12]. P73 belongs to the p53 family, and is directly involved in NSC proliferation; indeed, NSCs p73−/− shows a reduced size and a lower replicative rate [22]. This complex network is tightly regulated, and the precise mechanistic interaction is not completely clear in the embryonal development of the nervous system. Notch ligands themselves that are expressed from the neighboring cells exert an effect in this process. In particular, it has been shown that the Delta-like 1 intracellular domain (D1ICD), together with Numb, inhibits Notch signaling. D1ICD induces by lateral inhibition its differentiation effect in a Notch-independent manner by repressing the MAP kinase pathway through the inhibition of Erk1/2 phosphorylation and the induction of NSCs neuronal differentiation [23]. The Notch pathway also participates in stemness maintenance and senescence inhibition in cooperation with ataxia-telangiectasia mutated (ATM). Dong et al. isolated NSCs from ATM−/− mice and compared their characteristics with the wild type [24]. ATM-deficient NSCs reduced their replicative rate after three months of culture, as assessed by Ki67 analysis. ATM−/− NSCs, if induced to differentiate (1% Serum 7 dd), show a marked reduction in neuron numbers and an increased percentage of astrocytes [24]. Moreover, functional analysis has shown a decrease in Notch activity by approximately 40% in ATM−/− NSCs, and gene expression by RT PCR has revealed a decrease in the stemness factor Musashi [24]. Musashi is a translation regulator involved in neural development, and is involved in Notch/Numb pathways by regulating Numb translation [25]. RNA translation in NSCs is regulated not only by proteins, such as Musashi, but also by miRNA expression that influences cell fate. Among them, miR-9 has a pivotal role [26]: the down-modulation of its expression has a strong effect on neurosphere size, NSC proliferation, differentiation, and migration [27]. Like others, this miRNA shows multiple targets involved in neurogenesis, such as the monocyte chemotactic protein-induced protein 1 (MCPIP1), which regulates differentiation and migration [27]. MiR-9 also inhibits Hes-1 (Hairy and Enhancer of Split 1) expression, a Notch downstream gene [26,28]. In an opposite way, miR-138 and miR-485 expression reduces NSC proliferation by down-modulation of thyroid hormone receptor interacting protein 6 (TRIP6), a gene involved in NSC differentiation [29,30]. A distinctive feature of both miRNAs is the ability to induce NSC differentiation, despite their expression being reduced during the progression.

## 3. Brain Tumor Stem Cells

Two decades ago, it was found that the origin of a tumor mass in the brain is dependent on a specific cell named tumor-initiating cell (TIC) or BTSC. This led to the establishment of a hierarchical organization of cancer onset, in which BTSCs are responsible for self-renewal and metastatic spreading through symmetric division. Simultaneously, BTSCs can divide asymmetrically, differentiating into a cancer agglomerate. BTSCs most likely derive from a transformed NSC [6], and recent evidence indicates that glioblastoma originates from one of the regions where the NSCs are localized in adults, the SVZ [31]. In vitro experiments have demonstrated that after a prolonged culture NSCs undergo a spontaneous transformation into BTSCs [32]. After this identification, a number of studies investigated their characterization, starting with the specific markers, and their biology to discriminate them from their normal counterparts [33]. In fact, BTSCs share most characteristics with NSCs, starting with their ability to self-renew and differentiate, and their surface markers [33]. Both cell types have been found positive for Nestin or CD133 (prominin-1) [34] and grow in the same serum-free medium with bFGF and EGF. BTSC cellular markers have been studied for various reasons, one being the prognosis of glioblastoma; CD133 and high Nestin expression have been correlated with poor outcomes in glioblastoma patients [35,36]. CD133 expression also has a functional role, as demonstrated by in vitro and in vivo studies (recently reviewed by Ahmed et al. [37]), where glioblastoma stem cells (GSCs) with reduced CD133 showed a decreased sphere size due to the proliferation rate. Moreover, silencing CD133 expression in rat glioblastoma models results in an increased overall survival rate compared with the CD133-expressing tumors. Various research groups have analyzed functional markers in BTSC to narrow therapeutic approaches: among them a recent report identified integrin α-7 (ITGA7) as a key regulator in GSCs proliferation, and demonstrated its involvement in glioblastoma growth rate and invasiveness [38]. In particular, ITGA7 expression has been correlated with the laminin activation pathway, where ITGA7/laminin binding triggers focal adhesion kinase (FAK) and Src phosphorylation in GSCs [38]. Moreover, the evaluation of ITGA7 microarray data of glioblastoma patients from the Cancer Genome Atlas (TCGA) shows that the higher expression of ITGA7 correlates with a poor prognosis [38].

BTSCs show two additional notable features: the marked ability to migrate and thus enhance tumor invasiveness, and a high level of resistance to therapeutic treatments (radiation and/or chemotherapy). This last characteristic is made possible by the presence of sophisticated mechanisms enabling escape from DNA damage or apoptosis induction [4]. Several study groups have shown that numerous resistance mechanisms to therapy exist, including the inhibition of apoptotic response [4]. In particular, GSCs CD133+ show a substantial radiation resistance, while in xenograft animals treated with radiation (2 Gy) CD133+ showed an apoptotic response up to 5-fold lower compared to the CD133− counterpart [39]. Irradiation experiments have confirmed that DNA damage induces an increased survival through an elevated level of phosphorylation in the checkpoint proteins Rad17, CHK1, CHK2, and ATM [39]. Further research has been aimed at targeting ATM kinase for glioblastoma treatment. In particular, GSCs have been treated with KU-55933, an ATM kinase inhibitor, in association with radiation [40]. This combined treatment activates the DNA repair activity that protects these cells [39], inducing G2/M arrest and reducing cell viability [40]. It is remarkable that ATM signaling in NSCs induces senescence and proliferation reduction [24], and for this reason represents an attractive target for glioblastoma therapeutic approaches [40]. Another NSC key gene for stemness, Notch, has been implicated for some time in BTSC onset and maintenance [19]; in fact, its activation in association with K-Ras increases in glioblastoma mice models the percentage of Nestin-positive cells, indicating its direct involvement [41]. Notch activation, described above, activates Hes family transcription, where it has been demonstrated that Hes1 induces stemness in neuroblastoma BTSCs [42]. The overexpression of Hes1 results in a substantial enhancement of expression of the stemness markers (cKIT, Nestin, NANOG). Some experiments on neuroblastoma cancer stem cells (NBSCs) with Hes1 activated have shown, by limiting dilution, a stronger ability to form spheres compared with controls [42]. Moreover, serial transplantation analysis in xenotransplant models has shown an enhanced self-renewal ability due to the Hes1 transcription activity [42]. The pivotal role of the Notch pathway has also been assessed with the use of γ-secretase inhibitors, which impair its activation, reducing the symmetric division and undifferentiated progenitor maintenance of the NSCs [18,19], whereas it reduces proliferation and self-renewal ability in GSCs, suggesting that Notch inhibitors, such as γ-secretase inhibitors, could represent a promising application for glioblastoma therapeutic treatment [19]. Another class of Notch pathway proteins involved in its activation are the metalloproteinases ADAM10 and ADAM 17. These peptidase proteins have been specifically inhibited in BTSCs to increase migration ability, but induce differentiation [43], supporting the role of the Notch pathway for self-renewal and stemness maintenance, in BTSCs (and NSCs). In the same pathway, the contribution of Numb in BTSC cell fate and in brain tumor progression is important. In fact, Numb does not only act in opposition to Notch by inducing its degradation [44], but also, as described in the paragraph below, by modulating the stability of various proteins belonging to the p53 family, such as p53, p63, and p73, through the interaction with MDM2 and Itch [12,45,46]. Moreover, it has been found that Numb expression inversely correlates with glioblastoma prognosis [12]. In this context, another player is Musashi, which modulates Numb translation, among other functions [47]. Musashi overexpression increases GSCs sphere formation after multiple passages, increasing stem self-renewal [47]. Furthermore, Musashi expression is associated with glioblastoma radio-resistance by hyperactivation of DNA damage response (DDR) effectors after irradiation [47]. Beyond protein interplay, another common element among normal and tumoral brain SCs is represented by the miRNA profile. In fact, it has been found that they have comparable expression [48]. Some appear to be involved in resistance to temozolomide, one of the drugs used for glioblastoma treatment [49]. Analysis of miRNA in glioblastoma has been extensively carried out in an attempt to establish new therapeutic approaches (recently reviewed by Cheng et al. [50]), with perhaps the most interesting one belonging to the miR-34 family [51]. MiR-34a and miR34c have been shown to downmodulate Notch and Numb in GSCs. Their overexpression has been shown to induce inhibition of proliferation and cell death, and for this reason represents a suitable therapeutic approach [12,52].

## 4. Zika Virus in NSCs and BTSCs

The similarity between normal and tumoral neural stem cells offers a singular opportunity to merge knowledge obtained from the study of the Zika virus (ZIKV) and glioblastomas to improve therapies for neurodegenerative diseases and neoplasias. In this regard, a striking example is represented by ZIKV.

ZIKV is a member of the Flaviviridae family, and was isolated for the first time in the Zika forest in Uganda, in 1947, from a rhesus monkey [53]. The ZIKV genome is contained in an icosahedral capsid and consists of a positive-sense single-stranded RNA of about 10.7 kb [54]. The principal transmission vector is the Aedes mosquito. ZIKV infection in human adults is typically asymptomatic, with rare cases (<20%) of joint pain, rash, and mild fever that lasts for about a week. In exceptional cases it can induce some neurological complications, such as Guillain–Barré syndrome (GBS) [55]. The main impact of ZIKV is infection in pregnant women, in whom the virus also infects the fetus by crossing the placental barrier. At the fetal level, ZIKV can cause severe adverse effects, in particular neurodevelopmental disorder leading to microcephaly [56]. Studies conducted in vitro on NSCs and organoids have revealed that ZIKV impairs neural development by specifically targeting stem cells [57,58]. While neural undifferentiated cells appear susceptible to ZIKV infection, the differentiated counterpart seems less reactive [58]. Viral infection in undifferentiated cells induces a marked reduction in cell viability, with impairment of neurosphere formation [57,58]. ZIKV effects on NSCs or organoids induce a programmed cell death, supporting the effect observed in fetal neurodevelopment disorders. The selectivity of ZIKV for undifferentiated neural progenitors is at present still unclarified, though some experimental evidence has been offered. Various studies have proposed the tyrosine kinase receptor anexelekto (AXL), a member of the TAM (Tyro3, AXL, Mer) family, as the ZIKV entry point, while its inhibition reduces viral infection and NSC growth impairment [59,60,61]. Moreover, this receptor appears less expressed in neurons that are negligibly affected by ZIKV infection, and more expressed in astrocytes or glial cells that are strongly infected [61]. However, AXL is not the only receptor responsible for ZIKV entry, because its depletion does not preclude the infection [62]. Further evidence using proteomic approaches and validation by infection and gene targeting have shown that the virus infects neural progenitors through the neural cell adhesion molecule (NCAM1) receptor [63]. Other studies have identified, through CRISPR-Cas9 genome-wide screening, that viral infection takes place through the internalization of the integrin αvβ5 receptor (ITGB5) [64]. By down-modulating this receptor, they inhibited ZIKV infection [64]. The distribution of this last receptor exactly matches the cell tropism of the virus, because it is expressed mostly on the surface of the neural progenitor and glial cells, but not on neurons that are less susceptible to ZIKV infection [64]. Regarding the undifferentiated precursor tropism, a work by Ferraris and colleagues showed an increased differentiation of neural progenitors after ZIKV infection, and an induction of differentiation mediated by the Notch pathway [65]. Moreover, Notch pathway suppression using the DAPT inhibitor was shown to reduce the number of viral particles after infection [65]. Furthermore, Musashi inhibition and the alteration of the differentiation pathway modify ZIKV infection and the production of viral particles [66]. The link with the precursor-specific protein Musashi was further demonstrated by RNA pull-downs, where this protein directly interacts with the 3′UTR of ZIKV RNA and co-localizes with the replication intermediate (DS RNA), as assessed by confocal and STED super-resolution microscopy [67]. The same group showed that Musashi is essential for ZIKV replication, while its down-modulation resulted in a reduced number of viral particle production. A recent report defined the RNA binding region, demonstrating that it is conserved in the ZIKV strains, and it is not common to other flaviviruses [68]. The establishment of Musashi as a key protein in the replicative cycle of ZIKV clarifies the unique tropism of this virus for neural progenitors.

This unique tropism for neural undifferentiated progenitors prompted us and other groups to evaluate whether ZIKV can have an effect on BTSCs, considering the common features shared with NSCs. Consequently, once the impact on NSCs was demonstrated, the immediate step was to assess its influence on BTSCs, specifically focusing on GSCs [10]. GSCs express the same receptors as NSCs, in particular AXL [12], one of the receptors identified for viral entry in NSCs.

Supported by this evidence, we and others analyzed ZIKV activity on GSCs [10,12]. As expected, ZIKV infects human GSCs and inhibits their proliferation in vitro, inducing apoptosis, as evaluated by PI (Sub G0 fraction) analysis and caspase activation [10,12]. One intriguing aspect was the induction of the differentiation processes after ZIKV infection, as observed in NSCs [12,65]. The analysis of miRNA expression by NGS showed a clear increase in miR34c after ZIKV infection in all GSCs used [12]. This increase, confirmed by RT-PCR, is specific and directly dependent on ZIKV, which in turn modulates downstream targets of miR34c. In fact, we observed in GCSs infected with ZIKV, by Western blot analysis, a marked reduction in Notch, Bcl2 and Numb expressions [12]. The last appears to be either involved in the regulation of p73 expression in a proteasome-dependent manner [12]. Moreover, the overexpression of miR34c itself induces similar effects on cell proliferation and gene regulation [12]. Mir34 expression can be explained in different ways. One is the interferon (IFN)-mediated response induced by flavivirus and mediated by IRF3/Wnt, activated when cells are exposed to pathogen-associated molecular patterns (PAMPs) (e.g., viral double-stranded RNA) [69,70]. Another explanation is that ZIKV infection triggers DNA double-strand breaks (DSBs) and, consequently, p53 expression, which acts in a feedback loop with miR34, where p53 induces mi34 expression, and miR34 represses p53 [71,72].

Zhu et al. observed that more than 90% of GSCs infected cells were SOX2+ [10]. SOX2 is expressed in undifferentiated progenitor cells and is involved in neurodevelopmental and pluripotency processes [73]. The presence of SOX2 was further associated with an enhanced oncolytic effect promoting viral entry and ZIKV infection, in association with ITGA5 [73]. In particular, it was demonstrated that SOX2 promotes ITGA5 and enhances ZIKV infection [73]. GCSs xenograft treated with temozolomide (TMZ) and a ZIKV-attenuated strain demonstrated the efficacy of this approach in vivo for glioblastoma treatment [10].

## 5. Conclusions and Remarks

The study of stem cells is of paramount importance for regenerative medicine and the treatment of metabolic diseases. However, as ZIKV shows, there is no clear-cut path in science in which a virus involved in treating a developmental defect can contribute to a new therapeutic approach for such an extremely malignant disease as glioblastoma (Figure 4). The main effect of this virus in humans is the infection and the destruction of NSCs in embryos causing neurodevelopmental defects, whereas in adults the effects are very limited. Glioblastoma (grade IV glioma, WHO) is a nervous system malignant tumor, with a poor response to radiation/chemotherapy, and a median survival of about 14 months [74,75]. In most cases, the tumor recurs, with a fatal outcome [74]. New therapeutic options have been proposed, though no efficient treatment has been described. Cellular therapies using CAR (chimeric antigen receptor, CAR-T or CAR-NK) have been described [76,77]. Viral therapy herpes simplex virus 1 (HSV-1), adenovirus, vaccinia virus, reovirus, parvovirus, New Castle disease virus, and poliovirus do not appear to be successful approaches [11,78]. On the other hand, the encouraging effects of xenograft GBM mouse models infected with ZIKV, where the viral infection induces an oncolytic activity by reducing tumor size and metastatic diffusion, increasing animal survival, invite prompt translation to attempts at clinical therapies [10,79,80]. A single intracerebroventricular ZIKV dose has been shown to induce an oncolytic effect in vivo [80]. The same effect was also positively tested in immunocompetent dogs with spontaneous brain tumors, where ZIKV intrathecal injections significantly improved their neurological symptoms, extending their survival, with a reduction of tumor size and preservation of normal neurons [81]. However, 5 years after the first report [10], the only experience with ZIKV as a treatment in humans is a communication at a conference of a single compassionate case [82]. It is not clear why there has been such a delay in clinical trials for glioblastoma treatment with ZIKV, given that the virus alone has negligible effects in healthy humans. We hope that in the near future clinical trials or at least compassionate care will start for the treatment of glioblastomas with ZIKV.

## Figures and Tables

**Figure 1 ijms-24-13550-f001:**
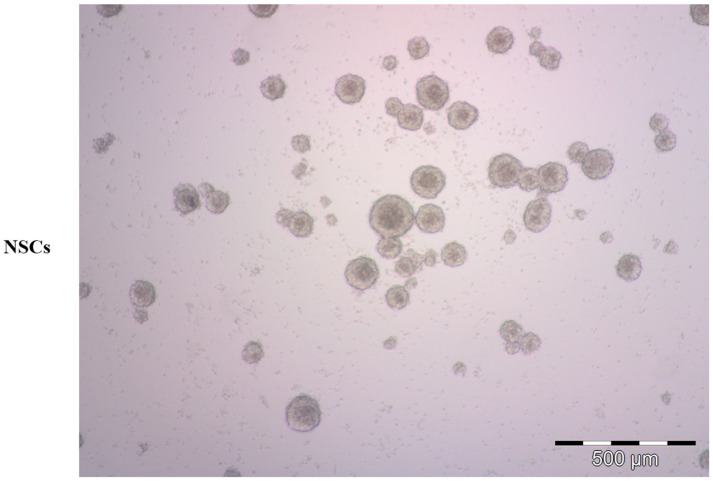
Representative bright-field microscopy image of human neural stem cells (NSCs). NSC-derived neurospheres were grown in serum-free condition, proliferating in suspension.

**Figure 2 ijms-24-13550-f002:**
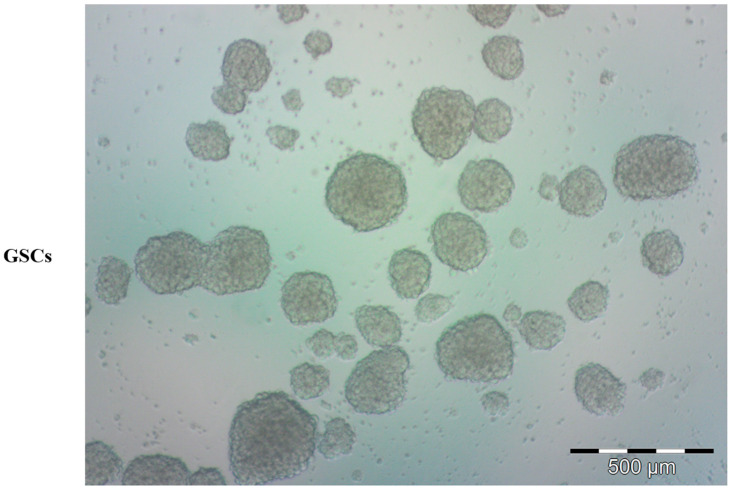
Representative bright-field microscopy image of human glioblastoma stem cells (GSCs). Tumor-derived spheres were grown in serum-free condition, proliferating in suspension.

**Figure 3 ijms-24-13550-f003:**
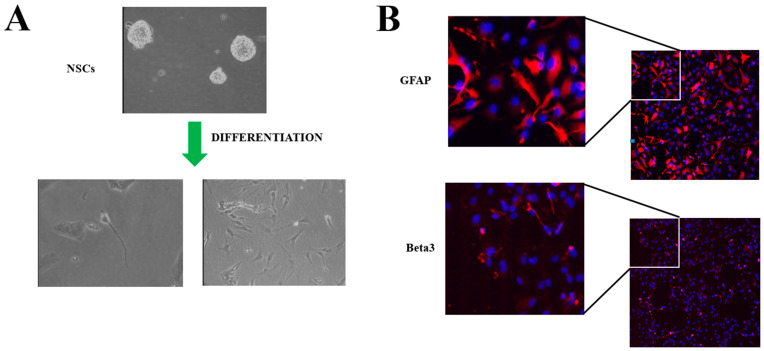
Human neural stem cells (NSCs): (**A**) representative bright-field microscopy image after differentiation by 2% FBS induction on Matrigel-treated plates; (**B**) immunofluorescence staining after differentiation of specific lineage markers. GFAP (glial) and Beta3 (neuron) in red, blue for nuclei (DAPI).

**Figure 4 ijms-24-13550-f004:**
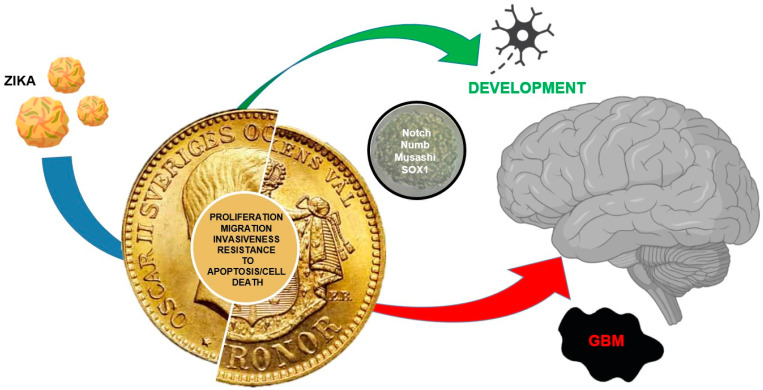
Two sides of the same coin. Development vs. cancer; both cell types display common features: proliferating ability, migrating ability/invasiveness, resistance to apoptosis/cell death. ZIKV that inhibits NSCs and induces cell death is involved in development and can be used against GBM because of its specific action toward GSCs (red arrow), inducing apoptosis and reducing tumor growth.

## Data Availability

Not applicable.

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
