# Peer review of "Two Sides of The Same Coin: Normal and Tumoral Stem Cells, The Relevance of In Vitro Models and Therapeutic Approaches: The Experience with Zika Virus in Nervous System Development and Glioblastoma Treatment"

_ijms, 2023, doi:10.3390/ijms241713550_

Round 1

Reviewer 1 Report

Neural stem cells (NSCs) have self-renewal capability and the differentiation potencial towards neural cell lineages. During the development, the normal and neoplastic NSCs have similar features, which might eithor contribute to the normal development or generate tumor in nervous system like brain tumor. Therefore, they can be considered as the two sides of the same coin, especially in the case of diseases infected by neuraltropic virus like Zika virus (ZIKV). This manuscript reviewed  the progress of NSCs and brain tumor stem cells (BTSCs) as well as ZIKV in NSCs and BTSCs, with illustrating figures. Overall, the manuscript was well organized and written in fluent English.

1. On page 3, Line 134-135: the authors described "recent evidence indicates that glioblastoma 134 originates from the same region where the NSCs are localized in the adult, the SVZ [31]". Actually, there are two main neurogenic sites which have NSCs in the brain, the subventricular zone (SVZ, this abbreviation needs to be defined) of the lateral ventricle and the subgranular zone (SGZ) of the dentate gyrus in the hippocampus.

2. Figures with neurosperes should be gaven more details in the legends: cultured how many days? "NSC" in Fig 3A should be replaced with "NSCs". "Gfap" in Fig 3B replaced with "GFAP". "Beta-3" or "Beta3 ???

Please check all these issues throughout the manuscript and keep these short terms consistent from the Introduction to the Conclusion.

Overall, the quality of English expression is good. But the authors still need to carefully check the manuscript thoroughly, because a few sentences needs to be improved. Such as " The isolation from adults and embryos was carried out by various laboratories and in different species, from mice to humans." in the Abstract. It's better to put "NSCs" to define "The isolation ---".

Author Response

  1. On page 3, Line 134-135: the authors described "recent evidence indicates that glioblastoma originates from the same region where the NSCs are localized in the adult, the SVZ [31]". Actually, there are two main neurogenic sites which have NSCs in the brain, the subventricular zone (SVZ, this abbreviation needs to be defined) of the lateral ventricle and the subgranular zone (SGZ) of the dentate gyrus in the hippocampus.

We thanks the reviewer for the accuracy, but the two zones and the abbreviation SVZ are already defined at line 40 in the Introduction.

“Despite there are two neurogenic zones, subventricular (SVZ) and subgranular (SGZ) zones, GSCs could originate from SVZ-NSCs, are more susceptible to oncogenic transformation than SGZ-NSCs (ref. Patterns of Invasive Growth in Malignant Gliomas-The Hippocampus Emerges as an Invasion-Spared Brain Region, Mughal et al. 2018, PMID: 29793116)”. We modified the text in the BTSC paragraph according to the request.

  1. Figures with neurospheres should be gaven more details in the legends: cultured how many days? "NSC" in Fig 3A should be replaced with "NSCs". "Gfap" in Fig 3B replaced with "GFAP"."Beta-3" or "Beta3 ???

Please check all these issues throughout the manuscript and keep these short terms consistent from the Introduction to the Conclusion.

We thanks the reviewer for noticing, we have made all the suggested changes.

  1. Overall, the quality of English expression is good. But the authors still need to carefully check the manuscript thoroughly, because a few sentences needs to be improved. Such as "The isolation from adults and embryos was carried out by various laboratories and in different species, from mice to humans." in the Abstract. It's better to put "NSCs" to define "The isolation ---".

We thanks the reviewer for the suggestions, we modified the sentence and checked all the manuscript.

Reviewer 2 Report

The authors have chosen a very interesting topic and non-standard approach for their review. The similarity of the metabolic pathways of neural stem cells and malignant nerve cells is one of the major biological questions, and its explanation may shed light on the nature of stemness. The author has conducted a comprehensive analysis of the published literature and reviewed the topics discussed above, but several objections must be considered before publication:

1. Closer reading and correction of the text is absolutely necessary. Pay attention to lines 40, 75, 267 and many others.

2. I propose to work on the figures:

- NSC and GBM cells should be present not only in neurospheres, but also in seeding cells. Thus, the less experienced reader will be able to get acquainted with their morphology.

- Fig. 3A - the authors need to indicate the line of cell differentiation.

- Fig. 3B - it is necessary to present higher magnification images of the cells in order to demonstrate the morphology of the differentiated cells.

-Figure 4 - Add a scheme explaining common metabolic pathways for NSC and BTSC, not just the idea of "double -face".

3. In the BTSC section of the manuscript, the cell types (GBM, NBM, etc.) should be listed as referenced, rather than generalizing them all as BTSC.

4. In Conclusion, pay more attention not only to the possibility of clinical trials, but also to the common/similar for NSC and BTSC mechanisms of response to a viral infection.

Author Response

REVIEW REPORT 2

The authors have chosen a very interesting topic and non-standard approach for their review. The similarity of the metabolic pathways of neural stem cells and malignant nerve cells is one of the major biological questions, and its explanation may shed light on the nature of stemness. The author has conducted a comprehensive analysis of the published literature and reviewed the topics discussed above, but several objections must be considered before publication:

  1. Closer reading and correction of the text is absolutely necessary. Pay attention to lines 40, 75, 267 and many others.

We thanks the reviewer for the suggestions, we modified the listed sentences and checked all the manuscript.

  1. I propose to work on the figures:
  2. a) NSC and GBM cells should be present not only in neurospheres, but also in seeding cells. Thus, the less experienced reader will be able to get acquainted with their morphology.

We showed the NSCs differentiation in figure 3A, moreover for a better comprehension we modified the text

  1. b) Fig. 3A - the authors need to indicate the line of cell differentiation.

As indicated below these are NSCs differentiated in 2% FBS medium, we modified the text for a better comprehension.

  1. c) Fig. 3B - it is necessary to present higher magnification images of the cells in order to demonstrate the morphology of the differentiated cells.

The figure was modified as requested.

  1. d) Figure 4 - Add a scheme explaining common metabolic pathways for NSC and BTSC, not just the idea of "double -face".

The figure was modified as requested.

  1. In the BTSC section of the manuscript, the cell types (GBM, NBM, etc.) should be listed as referenced, rather than generalizing them all as BTSC.

The text was modified as requested.

  1. In Conclusion, pay more attention not only to the possibility of clinical trials, but also to the common/similar for NSC and BTSC mechanisms of response to a viral infection.

We thanks the reviewer for pointing that, the text was modified as requested, however we tried to focus on the main final goal of our review is to convince the audience that is necessary a coordinate efforts to start clinical trial in the near future.